# Reproducibility report Generative causal explanations of black-box classifiers

## Reproducibility Summary

### Scope of Reproducibility

The paper by O'Shaughnessy et al. (2020) claims to have developed a method to disentangle the latent space of generative models during training. The latent space then consists of variables with causal influence and variables with non-causal influence. These can then be used as explanations of the generative model. These models will be reproduced with the goal of examining their latent space and confirming if they serve as sufficiently reliable explanations.

### Methodology

The github[1] of the paper contains a detailed README explaining how to reproduce the different figures. These steps were followed in order to reproduce the results. Additionally, an extension has been made by applying the method on a more complex dataset, namely ImageNet.

### Results

Generally speaking, the results in the paper are reproducible. The accuracy, however, when running Experiment 3 (38%) is much lower than in the paper. This is because we divided the amount of Monte-Carlo samples by 5. The difference between $\alpha$ and $\beta$ latent factors remains the same, even though the accuracy is much lower for $\alpha_1$ and $\alpha_2$. The results of the extension experiments did not show the same properties as in the paper. This, however, might be caused by factors other than the generalisability of the method in the paper.

### What was easy

The paper is clear and explains its concepts well. Also the provided code base and README file make it easy to reproduce the results of the paper.

### What was difficult

The GitHub of the paper is still being updated. Therefore one version might create similar results to the paper while an other one seems to result in errors. Also the code itself is not very well documented, which makes it difficult to solve the problem at such a low level.

### Communication with original authors

We have asked the authors for advice on our extension and they have provided useful information on how to approach our extension and why it has value beyond the original paper.

---

[1]https://github.com/siplab-gt/generative-causal-explanations

# 1    Introduction

Explainable Artificial Intelligence (XAI) refers to methods and techniques used to explain how a black-box classifier produces its outcomes. Interpretability is crucial to gain public trust in AI and machine learning. Post-hoc explanations approximate the behaviour of a black-box AI model by finding relationships between feature values and the classifier predictions. Causality in AI is another emerging research trend and it is seen as a major deficiency of current AI methods.

The paper describes a method for generating causal post-hoc explanations of any black-box AI model as part of explainable AI.

Main contributions of this study include a new framework for generating explanations, together with its analysis and evaluation on image recognition models.

# 2    Scope of reproducibility

The paper introduces a method for generating explanations of black box classifiers. This method is then implemented for Variatonal Auto Encoders (VAE) on the MNIST and Fashion-MNIST datasets with a standard CNN classifier. Several claims are made for their method:

- CLAIM 1: The latent space is split into two groups: $\alpha$ and $\beta$ latent factors. Out of these two classes, only $\alpha$ changes the class of the generated sample, while $\beta$ affects the sample without changing the class.
- CLAIM 2: The latent factors obtained with this method are effective at explaining classification decisions.
- CLAIM 3: The $\alpha$ latent factors contribute the most to the accuracy of the model.

In addition to examining these claims, an extension has been made on the work of the paper. The method for explaining black box classifiers has been tested for models that were trained on more complex data (e.g. ImageNet). This dataset is different from the MNIST and FMNIST datasets because the images are larger (64 by 64 pixels instead of 28 by 28), RGB instead of grayscale, and the images are less normalized. The images in MNIST and FMNIST are mostly centered and have a monochrome background color without any noise. Since these datasets are relatively easy and have fairly limited variety in their images, it is unclear how useful the explanations generated by this method would be on more complex datasets, therefore the main goal of the extension is to see if the method of the paper generalises for more realistic and noisy data as well as for a more complex generative model and classifier.

# 3    Methodology

The paper came with a Github repository[2] containing the source code and a detailed step-by-step explanation on how to reproduce the resulting figures. The figures have been reproduced, confirming the claims made by the authors in the paper.

- The first claim is checked on the MNIST dataset by reproducing figure 3 of the paper. This is done with the use of the python file *"make_fig3.py"*, which trains a VAE on the MNIST dataset for the classes 3 and 8. The CNN classifier can be trained separately by running the *"train_mnist_classifier.py"* file. This is also done for the FMNIST dataset in the file *"make_fig5cd.py"*.
- The second and third claim are checked by reproducing figure 5a of the paper. *"make_figure5ab.py"* Creates these figure as well as the figure for the first causal and non-causal factor of the Fashion-MNIST dataset VAE.

It should be noted that the authors used an NVIDIA's GeForce GTX 1080 GPU while we only had access to a NVIDIA's GeForce GTX 1050. As a result of this, the training of the models took longer and the number of samples for Monte-Carlo sampling was reduced. This may have affected the final accuracy visible in the figures, but the claims should still hold.

For checking these claims the models the paper suggested were used, namely a VAE and a CNN classifier on MNIST and Fashion-MNIST. Next we will make an extension on the paper by using the same model with more complex images, using images from the ImageNet dataset. The dataset will be limited to two classes: horses and zebras. This subset of ImageNet was found on Kaggle[3]. Using two classes that are already similar should make it easier to train the model

---

[2]https://github.com/siplab-gt/generative-causal-explanations
[3]https://www.kaggle.com/suyashdamle/cyclegan?select=horse2zebra

while still retaining the desired attributes of a disentangled latent space capable of explaining classifier decisions. This is done similarly to the process in the paper where only two digits of MNIST are used that look similar. We expect to see a causal latent factor that either determines whether the generated image has stripes or not, or a causal latent factor that affects the color of the animal in the pictures.

### 3.1 Model descriptions

Two models were used to create each of the figures: A VAE and a CNN classifier. The VAE structure itself was left untouched and can be found in the appendix of the original paper.

For figure 3, a VAE was trained on the MNIST dataset for the digits 3 and 8. The VAE was then used to generate samples. The VAE model was trained with one causal factor and seven non-causal factors (denoted in the paper as K=1 and L=7, respectively). The model was trained for 8000 training steps with a lambda of 0.05, a batch size of 64 and a learning rate of $5e-4$. The model in the paper used 25 Monte-Carlo samples for the causal factors and 100 samples for the non-causal factors. For the reproduction, this was divided by 5 as this was the maximum amount of samples that fitted on the GPU without causing a memory error. 5 samples were used for the causal factors and 20 samples were used for the non-causal factors.
A CNN classifier for the MNIST dataset was trained for the digits 3 and 8 as well. The dataset parameter was set to *mnist*, the class use to *[3,8]*, the batch size to 64, the c_dim to 1, the learning rate to 0.1, the momentum to 0.5, the img_size to 28, the gamma to 0.7 and the number of epochs to 50.

For figure 5, another VAE was trained, but this time it was trained on the Fashion-MNIST dataset. Only 2 causal and 4 non-causal factors were used for this model (K=2, L=4). The used classes were also changed to *[0,3,4]*, indicating the shirt, dress and coat classes of the dataset. The Monte-Carlo samples were reduced in the same way as before. The rest of the parameters were also the same as before. Similarly, the CNN classifier used mostly the same parameters. The only parameters that were changed were the dataset that was changed to *fmnist* and the class_use was changed to *[0,3,4]*. "make_fig5cd.py" Uses the exact same models as figure 5 a and b.

The model used for the ImageNet images is also a VAE. The structure of a VAE trained on images of the CelebA dataset was used[4] as this dataset is similar to the dataset we use: the images are RGB and the image sizes are 64 by 64 pixels. By using an already existing structure, a minimal amount of time is spent looking for a model capable of representing and generating images for our ImageNet dataset. This model is more complex than the model previously used for the MNIST and Fashion-MNIST datasets. This is because the structure of the previous VAE proved to be too simple to represent the ImageNet data, resulting in poor generated samples.

A new classifier also needs to be trained as the imageNet data is new and more complex. This classifier is not the main importance of the extension as the goal is to be able to explain the behaviour of this black-box classifier, not obtain an optimal classifier. To stay close to the original paper a CNN-classifier was trained closely matching the classifier used in the original paper. The first difference is the input channels, which goes from 1 to 3 to accommodate the RGB color images. The image size also warrants a change in the fully connected layers input as there is more data input so the input size of the first fully connected layer becomes 57600 where it was 9216 in the original paper using MNIST or Fashion-MNIST. The last detail changed when training the CNN-classifier is the learning rate which became 0.001 after some tweaking. This resulted in an accuracy of 0.775 on the test set after 15 epochs.

Finally, a figure similar to figure 3 of the paper is created in *make_figextension.py*. The previously used code for figure 3 is also partially reused. This file includes: loading the data, loading the VAE and classifier models, training the gce and plotting the images in a grid. In *make_figextension.py*, the classifier and VAE classes were changed accordingly and a new function for plotting the final figure was added in *plotting.py* in order to handle RGB images. The parameters used for creating these figures were: Dataclasses = *[0, 1]*, indicating the horse and zebra class, K = 3, L = 21, 8000 training steps, Nalpha = 3, Nbeta = 12, lambda = 0.05, a batch size of 16 and a learning rate of $5e-4$.

Before extending an existing implementation, one should answer the following questions:

Q: Why are the current experiments not good enough for generalisation?
A: The datasets used are not very complex. For instance, MNIST database contains handwritten digits only. The size of the images is fairly small at 28 by 28 pixels. The images are also normalised, the objects are mostly centered, oriented uniformly and the backgrounds do not contain noise. This means that the current experiments only show that it generalizes well on standardised and normalised data but not on natural images such as those of ImageNet

---

[4]https://github.com/yzwxx/vae-celebA/blob/master/model_vae.py

124 Q: What conclusions could we possibly draw by using this extension?
125 A: The results will tell us if this method generalizes well on more natural images. It also shows how the need for a
126 larger model with a higher capacity grows as the sample complexity and size increases.

127 Q: What can we learn from this?
128 A: How to test the generalisation of a model.

## 3.2 Datasets

130 Two datasets were used for reproducing the paper: The Modified National Institute of Standards and Technology dataset
131 (MNIST) and the Fashion-MNIST dataset.
132 The MNIST dataset was split into a training set of 50 000 images, a validation set of 10 000 images and a test set of 10
133 000 images. Only the used classes were extracted from these sets.
134 The Fashion-MNIST dataset was split into a training set of 60 000 images, a validation set of 6 000 and a testing set of
135 4 000 images. Similarly, only the relevant classes were extracted from these sets.

136 In Fashion-MNIST only the shape must be detected. The details don't really matter.
137 The ImageNet dataset contains a large amount of images of different sizes. A decision was made to only use images
138 that belong to the *horse* and *zebra* classes. This is similar to how only two or three classes are used for MNIST and
139 Fashion-MNIST. In addition to this, only images of 64 by 64 pixels will be used. The original ImageNet dataset that was
140 used contained images of 128 by 128 pixels. The code in *resize_horse2zebra.py* downsizes the images of the ImageNet
141 dataset and saves them under *horse2zebra64*. This python file uses the default parameters of the resize function from the
142 PILL Image library[5]. ImageNet contains even larger images. However, 64 by 64 strikes a good balance in complexity
143 without increasing the training time too drastically or requiring an even more complex model.
144 After having downsized the images, a new dataloader is used to load the ImageNet images as a numpy array. This is done
145 using the code in *load_horse2zebra.py*. Due to the differences between this dataset and the MNIST/Fashion-MNIST
146 dataset, the file to load the data had to be made from scratch. The ImageNet dataset contained a few grayscale images,
147 which were left out for consistency. The final dataset consisted of 2661 images, of which there were 1067 training and
148 120 testing images of horses and 1334 training and 140 testing images of zebras.

## 3.3 Experimental setup and code

150 The source code ran without any problems after changing the Monte-Carlo sampling parameters as described in the
151 Model descriptions section. There was only a single exception inside the *"train_mnist_classifier.py"* file where the
152 validation was originally done without disabling the gradients, resulting in the code using too much memory. This was
153 solved by adding *with torch.no_grad():* right before the validation loop. This is only a problem if a GPU with a small
154 amount of memory, such as a GTX 1050 with 2GB of memory, is used. The code of this report can be found on the
155 GitHub page[6].

## 3.4 Computational requirements

157 The hardware used were an NVIDIA GTX 1050 with 2GB of memory, an Intel Core i7-7700HQ and 12 GB of DDR4
158 RAM. The training of the CNN classifiers took roughly 4 minutes for the MNIST and FMNIST datasets. The GCEs
159 took slgihtly more than 8 minutes to train from scratch. For the extension, the classifier also took about 4 minutes to
160 train. The reason for this is because the classifier converged before 15 epochs. Finally, the GCE took 30 minutes to
161 train. This was to be expected as the VAE used for this experiment was more complex than those used for the MNIST
162 and Fashion-MNIST datasets.

## 4 Results

164 From the code of the authors we were able to reproduce very similar results to those of the paper. By following the
165 steps in methodology we were able to reproduce the figures of the paper with only minor differences. In the paper some
166 of the results are left out that are generated by following the steps of the methodology of the paper. In this section we
167 will show all the generated results to further explore the claims of the paper. These extra results are often the extra
168 non-causal latent factors that contribute to representing the data distribution.

---

[5]https://pillow.readthedocs.io/en/stable/reference/Image.html

[6]https://github.com/UvAartificialintelligence/Fairness-Accountability-Confidentiality-and-Transparency-in-AI

### 4.1 Results reproducing original paper

#### Experiment 1

The results of experiment 1 are the recreation of figure 3 of the paper. In the images the coloured boxes indicate the classification of our classifier. Here a blue box indicates the classification of the number 3 and a yellow box indicates the classification of the number 8. This experiment supports claim 1 and 2. looking at figure 1 and 2 we see that $\alpha$ changes the classification of the classifier while $\beta$ shows change in non-causal latent factors but not the classification of the classifier. It is not always clear what latent factor of $\beta$ is but it is clear to see that factors like thickness and rotation are defined by the latent factors in the figures 2:8. The results of this experiment are nearly identical to that of the paper. Figures 5:8 show the remaining results that are not in the paper. These figures also support claim 2.

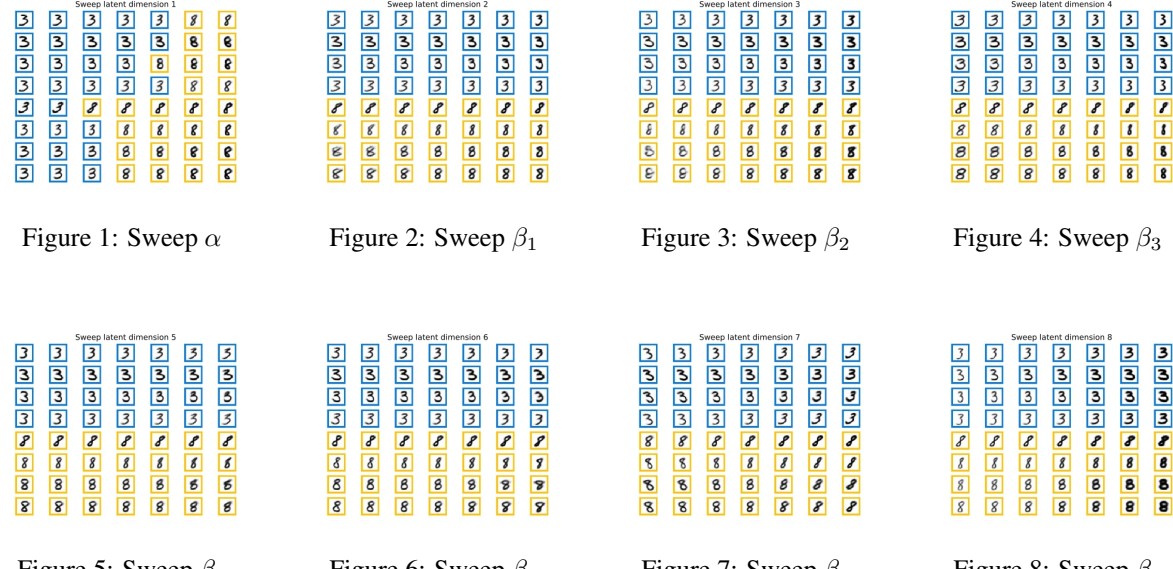

Figure 1: Sweep $\alpha$      Figure 2: Sweep $\beta_1$      Figure 3: Sweep $\beta_2$      Figure 4: Sweep $\beta_3$

Figure 5: Sweep $\beta_4$      Figure 6: Sweep $\beta_5$      Figure 7: Sweep $\beta_6$      Figure 8: Sweep $\beta_7$

#### Experiment 2

The results of experiment 2 are the recreation of figure 4 of the paper. This experiment supports claim 1. This experiment was used to show how the method would compare to popular explanation models. In the paper the result is much more compact to better show the comparison to the other models. Yet in figures 9 and 10 we can see that the $\alpha$ is able to control the class and the $\beta$ controls the non-causal factors of the images.

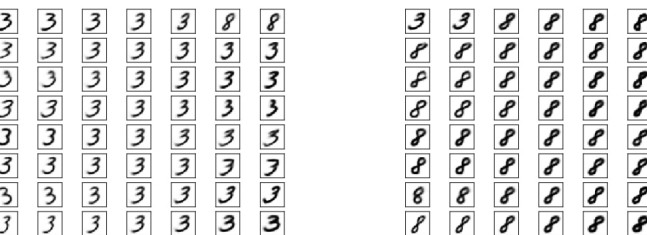

Figure 9: comparison of global      Figure 10: comparison of global
explanations with other methods      explanations with other methods

#### Experiment 3

The results of experiment 3 are the recreation of figure 3 of the paper. In figure 11 the information flow of each latent variable is shown in a bar diagram. Here we see that the causal effects indeed contain the most information about our images. In figure 12 we see a how removing a latent factor influences the accuracy of the model. When removing one

of the causal latent factors there is a significant drop in accuracy compared to removing one of the non-causal latent factors. The re-encoded accuracy of our model stands at 85% which is slightly higher than that of the original paper at 81%. Also the accuracy of our $alpha_1$ is also slightly higher at 55% instead of 50%. In figures13:18 we see the influence of the latent factors on the classification. This experiment supports claim 1, 2 and 3. The accuracy's are lower than in the original paper. This is because we divided the amount of Monte-Carlo samples by 5. The difference between $\alpha$ and $\beta$ latent factors remains the same, even though the accuracy is much lower for $\alpha_1$ and $\alpha_2$. Additionally, the code for generating this image does not have a seed, meaning the results are different every time the figure is created. Aside from these difference, it also seems that all $\beta$ latent factors have the same effect on the accuracy of the model. This further supports claims 1 and 3. In figures 15:18 we again see that the latent factors capture the small variations of the images but are not able to change the class of the images. This further supports claims 1 and 2. Our results show two causal latent factors and four non-causal latent factors while in the paper only one causal factors and two non-causal factors are shown.

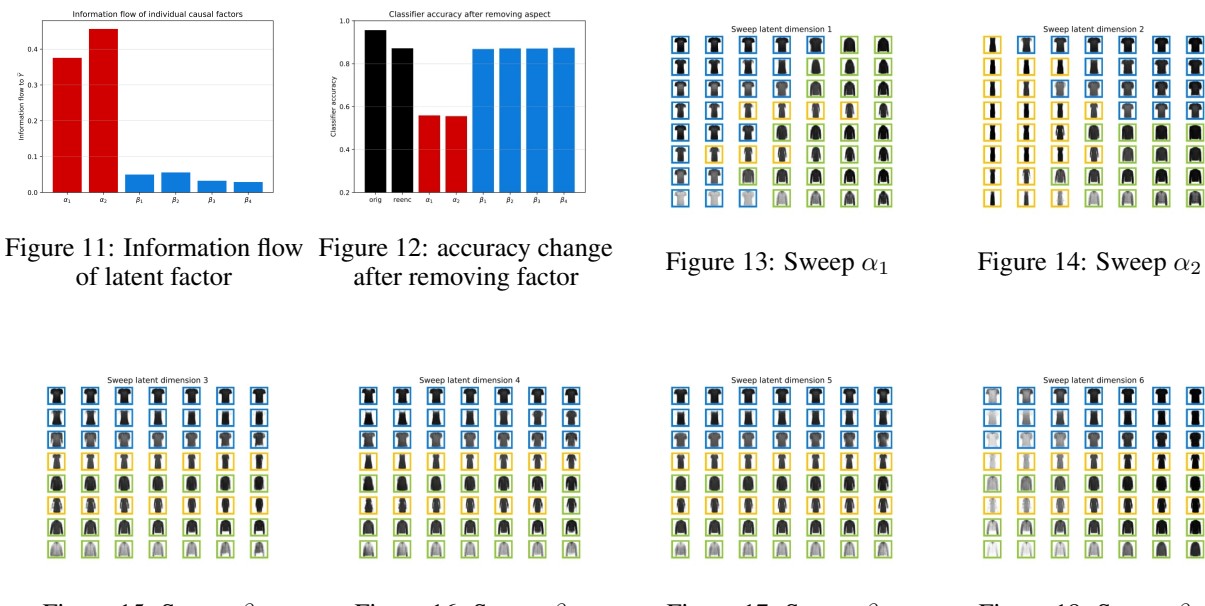

Figure 11: Information flow of latent factor

Figure 12: accuracy change after removing factor

Figure 13: Sweep $\alpha_1$

Figure 14: Sweep $\alpha_2$

Figure 15: Sweep $\beta_1$

Figure 16: Sweep $\beta_2$

Figure 17: Sweep $\beta_3$

Figure 18: Sweep $\beta_4$

## 4.2 Results extention

Experiment 4

The results of experiment 4 show the generated samples of the VAE in a similar fashion as for experiment 1. A blue border around the image indicates that it was classified as a *horse*, while an orange border indicates that it was classified as a *zebra*. The results of two different models are shown. The first results show all three causal factors and one non-causal factor of a VAE trained with a lambda of 0.05. These can be seen in figures 19:22. The second set of results show the same results but for a VAE trained with a lambda of 0.5. These can be seen in figures 23:26. A lower lambda should result in a stronger causal relation between $\alpha$ and the classification. As can be seen in the results, this was not the case for the more complex VAE trained on ImageNet. There is no real consistency between the assigned class of the samples when moving over any of the latent factors. As a result of this, there is also no consistent difference between the causal and non-causal results.

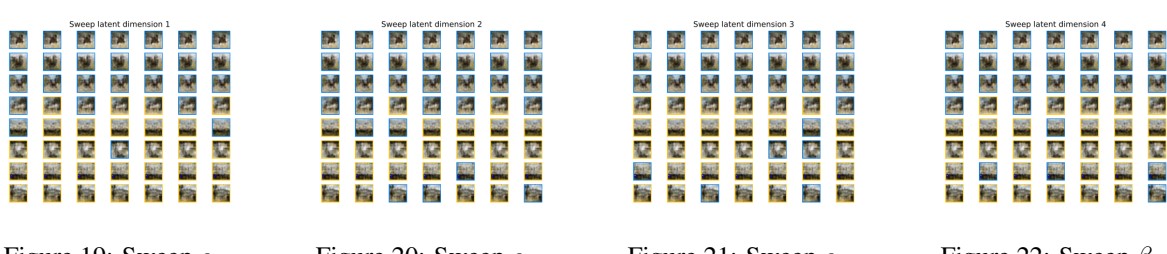

Figure 19: Sweep $\alpha_1$

Figure 20: Sweep $\alpha_2$

Figure 21: Sweep $\alpha_3$

Figure 22: Sweep $\beta_1$

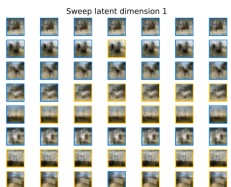
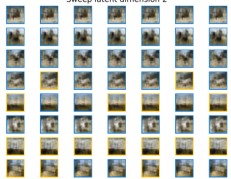
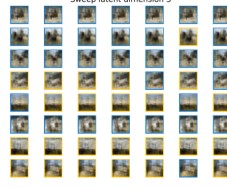
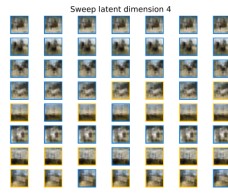

Figure 23: Sweep $\alpha_1$  Figure 24: Sweep $\alpha_2$  Figure 25: Sweep $\alpha_3$  Figure 26: Sweep $\beta_1$

## 5  Discussion

The claims of the paper were supported by the results in our experiments. We managed to reproduce all the steps described in the GitHub. Even though the amount of Monte-Carlo steps was reduced, the same behaviour could still be seen in the results.

We also performed additional experiments to check the generalisability of the method. The results of the fourth experiment did not show the same behaviour as the experiments in the paper. It is most probable that this is not due to the fact that the method in the paper does not generalise well. The model used for the VAE was not extensively explored due to lack of time. The structure used for the VAE was created for a dataset containing faces. This dataset contains less noise than our dataset and appears to be overall more normalised. The subset of the ImageNet dataset was also fairly small in comparison to the MNIST and Fashion-MNIST dataset, especially when taking in mind that more complex data often also means more training data is needed. In addition to this, the ImageNet images also contained more noise. This resulted in the classifier underperforming. In future work it would be best to start of with the classifier. Once a classifier is acquired that can consistently distinguish the two classes of the dataset, a potential better VAE can be searched.

### 5.1  What was easy

The paper is clear and explains its concepts well. In order to understand the paper, one does not need a lot of prior knowledge about the broader concepts such as causality.

The code base of the paper is easily accessible in GitHub via a link in the paper. A README file on the GitHub contains step by step instructions on how to reproduce the results found in the paper. This file also contains the parameters that need to be changed for every experiment.

### 5.2  What was difficult

In the beginning it was not possible to reproduce any figure with the given files and parameters from the GitHub. Furthermore there was no extensive documentation in the code or an explanation on what the parameters were. There are a lot of python files that are all connected. This made understanding the models quite a challenge. The GitHub was changed after a few weeks in the project. It then worked with the given parameters. Since then we have had no more problems running the code except for some hardware-related memory errors. These were, however, easily fixed.

These problems could have been avoided if more extensive documentation existed on how the files worked. Also, the code itself was not commented properly. For example, not all parameters of the files are explained.

### 5.3  Communication with original authors

An e-mail was sent asking whether the authors had already trained an explainable VAE on an ImageNet classifier. They did start adding some code to use ImageNet, but they never actually tested it out. In fact, they did not end up testing on anything larger than MNIST/Fashion-MNIST. The authors also recommended first training a VAE without causal effect in order to make sure the model and dataset worked.

## References

O'Shaughnessy, M., Canal, G., Connor, M., Davenport, M., and Rozell, C. (2020). Generative causal explanations of black-box classifiers.

