# OpenReview forum: "[Reproducibility Report] Generative causal explanations of black-box classifiers"
_ML_Reproducibility_Challenge/2020 — Reject_

### Official Review · AnonReviewer2 · 2021-02-27
**A thorough reproducibility study with author provided code.**

**Rating:** 7
**Confidence:** 3

**Review:**

The paper provides a thorough explanation of the reproducibility efforts. It could possibly be improved by providing more experiments, especially on the attempts to apply the approach to new examples. (Hyper)parameter choices are mostly adopted from the paper and could have been experimented with, too.

The summary is somewhat vague in some places. For instance, the authors state: "The accuracy, however, when running Experiment 3 (38%) is much lower than in the paper. This is because we divided the amount of Monte-Carlo samples by 5" This needs some motivation as to why the number of samples was changed and whether this is a problem or not.

Great to see, authors have been in touch with the original paper authors.

**Familiar With The Original Paper:**

I have read the original paper

**Reproducibility Summary:**

Report has summary

---

### Official Review · AnonReviewer1 · 2021-03-04
**Review of [Reproducibility Report] Generative causal explanations of black-box classifiers**

**Rating:** 5
**Confidence:** 3

**Review:**

This work aims to reproduce and examine the claims made in "Generative Causal Explanations of Black Box Classifiers". The authors do so by reproducing the original results presented in the paper and additionally evaluating against imagenet to test the behavior on a more challenging dataset. The authors remark that they are unable to fully reproduce the results, though this is likely because they did not evaluate using the exact same procedure due to a reduction in the number of Monte Carlo samples. They also evaluate using the image net dataset and show worse behavior. However, it isn't clear to me if this degraded performance can also be attributed to the same issues that prevented reproducibility on the original experiments.

Overall, I think the authors made a nice attempt at reproduction (MC samples aside), and do well to consider an additional dataset. I found the rationale behind the extension to be solid, and very important when thinking through considerations that go into applying a model in more realistic settings.

I have two core issues with this reproduction:
1. The paper is generally difficult to follow. The paper reads closer to an outline than a finished report. I would encourage the authors to spend some additional time on organization, making sure that the key takeaways are made plain and that the report reads fluidly throughout.
2. This reproduction tests one additional dataset is commendable, but I would have liked to see some examination that gets closer to the behavior of the method. For example, robustness to hyperparameters, model complexity, and other behaviors.
3. It is not entirely clear to me why the authors chose to use less MC samples, since it hampered the ability to fully evaluate the claims of the paper.

**Familiar With The Original Paper:**

I have read the original paper

**Reproducibility Summary:**

Report has summary

---

### Decision · Program_Chairs · 2021-03-31

**Decision:**

Reject

**Comment:**

Overall reviews and/or the paper content not good enough for the AC to recommend to the journal.